# OpenReview forum: "Recontextualization Mitigates Specification Gaming Without Modifying the Specification"
_ICML.cc/2026/Conference — ICML 2026 regular_

### Official Review · Reviewer_18KG · 2026-03-12

**Soundness:** 3
**Presentation:** 4
**Significance:** 3
**Originality:** 3
**Overall Recommendation:** 5
**Confidence:** 4

**Summary:**

This paper proposes recontextualization, an off-policy training approach for mitigating specification gaming in language models. The core idea is to modify training data by conditioning on prompts that are *more permissive* of specification gaming than those used to generate those data in the first place. This leads to training settings where (1) good behavior is rewarded in spite of misbehavior-permissive instructions, and (2) misbehavior is only ever reinforced conditional on permissive instructions. Empirically, downstream models are less likely to misbehave at inference-time. This is demonstrated in four sets of experiments, where recontextualization is shown to achieve a better pareto-frontier on desired vs undesired behavior when compared to alternative techniques like inoculation prompting, penalty/screening-based interventions, and KL regularization.

**Compliance With Llm Reviewing Policy:**

Affirmed.

**Final Justification:**

The authors' rebuttal addressed my concerns well, so I have increased my final score from 4 --> 5.

**Key Questions For Authors:**

1. Given the inconsistent results across settings (generic prompts working best in Section4.1, loophole-specific prompts in Section 4.3, inoculation partially working for sycophancy but not elsewhere), what practical guidance would the authors offer a researcher choosing a recontextualization prompt for a new setting? Have the authors considered testing what happens when the hypothesis about the misbehavior is wrong? This would help me better understand the practical applicability of the method.
2. In production post-training, multiple failure modes may be targeted simultaneously. Could the authors test the application of recontextualization for two or more misbehaviors at once? Given the observed IFEval degradation from a single recontextualization, understanding whether these effects compound is important, as this would lower my confidence in the approach.
3. What happens if recontextualization is applied in an environment where exploits are particularly subtle or do not exist at all? Specifically, could the misbehavior-permissive training prompt increase the salience of potential exploits the model wouldn't otherwise have discovered? It would be reassuring if this potential side-effect was an irrelevant concern.
4. The irrelevant recontextualization results in Appendix F.6 show that some semantically unrelated prompts partially reduce reward hacking in the Leetcode setting. How do the authors reconcile this with the claim that prompt semantics drive the effect? Even a speculative explanation here would, in my opinion, strengthen the paper’s contribution.

**Limitations:**

yes

**Strengths And Weaknesses:**

### Strengths

- I find the core idea of this work very interesting! It is well-differentiated from prior methods like context distillation and even from concurrent approaches like inoculation prompting. This is articulated clearly in the paper (generation data vs training data). The importance sampling argument in Appendix A, explaining that the method works *because* of its bias, is also appreciated.
- The problem is important and timely; specification gaming is a real concern in production post-training, and this mitigation seems practically valuable, due to its relative simplicity in implementation. This increases its likelihood for adoption.
- The paper varies experimental details (e.g., models, learning algorithms) in four qualitatively different settings, includes and controls for several baselines and alternatives (e.g., inoculation prompting), and is honest about the limitations of the method (e.g., instruction-following concerns, off-policy data, problems with adapting to reasoning models).
- The paper is also easy to follow and well-structured; I particularly appreciated the A → B notation.

### Weaknesses

- **Experiments lead to somewhat conflicting results.** In Section 4.1, generic prompts outperform task-specific ones, whereas in Section 4.3, only prompts that explicitly describe the loophole work best. In the sycophancy setting, inoculation prompting partially works, to a greater extent than in other settings. These differing results make it hard to know what kind of recontextualization prompt to actually use in a new setting. This limits the practical usability of the method.
- **Realism of experimental setups is marginal.** All four setups deliberately inject or construct the misspecification that recontextualization then mitigates, and reward signals are deliberately designed to assign high scores to specification gaming. In practice, the whole problem with specification gaming is that researchers don’t know how the reward is misspecified. While it is certainly useful to construct these setups in this way for the sake of clear experiments, I would have liked to see more stress-testing of the assumption that researchers can hypothesize the misbehavior when it is unknown.
- **Instruction-following degradation seems underplayed.** IFEval drops appear across multiple experiments, and Appendix C shows models become less responsive to adversarial evaluation prompts. Instruction-following is itself a critical component of alignment, and it is unclear whether these issues could compound if, for example, multiple recontextualizations for different misbehaviors were stacked simultaneously, as could be the case in a production environment. This is my largest concern with this approach, so I would have appreciated a deeper investigation into it.
- **Issues with adapting the approach to reasoning models.** The paper’s framing is explicitly about RL, which is where reasoning model training currently occurs. But, as acknowledged in the limitations, reasoning models are likely to verbalize instructions in their CoT, which could conflict with the recontextualization training prompt. This is a significant gap given the framing.
- **Discussion of the recontextualization mechanism is light.** We get two intuitions (resistance learning and conditional containment of misbehavior), but no formalization. The Appendix A result tells us the bias is necessary, but not what it's doing. The paper would benefit from even speculative analysis of what distinguishes recontextualization from e.g., inoculation prompting mechanistically, and why certain prompts work in certain settings.

---

> ### Author Rebuttal · Authors · 2026-03-31
>
> Thank you for the thorough review, deep engagement with the work, and constructive criticism. We believe your comments will be highly valuable for future work on this topic.
> ### Prompt sensitivity and practical guidance
> We agree that variation across settings makes prompt selection an important practical question. Our current takeaway is that recontextualization works best when the prompt clearly targets the class of behavior one wants to suppress, though the required specificity can vary by setting.
> Our practical recommendation is to use a prompt that directly targets a suspected exploit, e.g., "prioritize user satisfaction, even if that means being sycophantic" or "exploit vulnerabilities in the testing scheme." One possible explanation for the discrepancy between Section 4.1 (generic prompts suffice) and Section 4.3 (only specific prompts work) is that smaller models (Qwen3-4B) are less responsive to vague instructions than larger ones (GPT-4o-mini), and may require more explicit framing of the failure mode. This tentatively suggests defaulting to more specific prompts with smaller models.
> When the exploit is hard to anticipate, we recommend high-level prompts such as "reward hack whenever you get the opportunity," following MacDiarmid et al. Better prompt-selection heuristics remain an active area of investigation.
> ### Multiple misbehaviors
> We appreciate the concern and now highlight instruction-following degradation earlier in the paper. As discussed in the Limitations and Appendix C, some degradation is expected: recontextualization reduces misbehavior in response to adversarial instructions, reflecting a natural tradeoff between helpfulness and harmlessness.
> Testing recontextualization against multiple misbehaviors simultaneously would be a valuable extension, understanding whether IFEval degradation compounds is particularly important. We are also interested in comparing the effectiveness of individual targeted prompts against a single general prompt in such settings.
> ### Surfacing exploits
> We note that in MacDiarmid et al., the reward hacking prompt leads to earlier onset of reward hacking. However, in recontextualization the prompt is added after generation and therefore cannot directly cause the model to explore undesirable solutions during rollouts, and the model is not trained on the tokens describing the exploit.
> We acknowledge that training the model to resist certain instructions could make them more salient via some currently unknown mechanism. However, given our results and the findings of Wichers et al. and Tan et al. showing that including unsafe prompts during training makes models safer, we currently consider this an unlikely side effect.
> ### Irrelevant recontextualizations
> One possible explanation is a conditionalization effect analogous to that proposed for inoculation prompting (lesswrong.com/posts/znW7FmyF2HX9x29rA). Misbehavior becomes conditioned on the irrelevant prompt, functioning as a backdoor. The fact that coding performance on neutral prompts also improves suggests that capabilities might generalize more broadly than propensities: the ability to solve a LeetCode problem transfers across prompt contexts, whereas the propensity to reward hack becomes more tightly bound to the specific training prompt.
> ### Experimental setups
> We agree our experiments rely on artificially injected misspecifications. However, our recontextualization prompts reasonably simulate plausible developer knowledge: a developer using test-case-based evaluation could anticipate test-case hacking; sycophancy during post-training is well-documented; and so on. Our results in Sections 4.1 and 4.3 also suggest that large models respond to vaguer prompts, reducing the specificity required in practice.
> What remains untested is whether recontextualization holds up when the developer's hypothesis is significantly off. The vague-prompt results suggest some robustness to partially wrong hypotheses, and a misspecified prompt may simply act as a semantically irrelevant recontextualization, though we acknowledge this deserves direct testing and have added it to the Limitations section.
> ### Reasoning models
> Section 4.2 suggests recontextualization can still be effective when the model verbalizes instructions in its chain of thought (Table 15). We agree future work should more thoroughly study the interaction between conflicting reasoning content and training prompts, and note this concern is not unique to recontextualization: Schoen et al. show that deliberative alignment can lead to "blatant miscitation" when specifications are removed from context.
> ### Mechanistic understanding
> As discussed with Reviewer qsat, this reflects the current state of the broader literature on both recontextualization and inoculation prompting. What distinguishes the two approaches mechanistically, and why certain prompts succeed in certain settings, remain important open questions we hope future work will address.

---

> > ### Author Rebuttal · Reviewer_18KG · 2026-04-02
> >
> > Thank you for your thorough and honest responses to my questions - I will increase my score. I encourage incorporating the practical guidance and discussion into the final version.

---

### Official Review · Reviewer_5Dov · 2026-03-13

**Soundness:** 3
**Presentation:** 3
**Significance:** 3
**Originality:** 3
**Overall Recommendation:** 4
**Confidence:** 2

**Summary:**

This paper introduces recontextualization, a simple on-policy training intervention designed to mitigate specification gaming without modifying the underlying (potentially misspecified) supervision signal. The method first samples model completions under prompts that discourage a target misbehavior, and then trains the model on the same completions paired with prompts that permit or encourage that behavior. In doing so, the model learns to avoid the misbehavior even when it is seemingly allowed. Across four settings,revealed-metric gaming in natural language tasks, test-case hacking in code generation, evaluator-overwriting in code RL, and sycophancy during post-training, the authors demonstrate that recontextualization consistently reduces gaming behaviors while maintaining or even improving task performance.

**Compliance With Llm Reviewing Policy:**

Affirmed.

**Final Justification:**

The authors’ rebuttal addressed most of my major concerns. As a result, I have increased my score from 3 to 4.

**Key Questions For Authors:**

see above

**Limitations:**

Yes

**Strengths And Weaknesses:**

Strengths:
- Recontextualization is a clean, general mechanism that contrasts data-generation and training contexts to shape generalization.
- The approach requires only natural-language prompt changes and integrates with standard on-policy loops.

Weaknesses:
- Heavy reliance on LLM judges for scoring quality and sycophancy.
- Some experiments have small numbers of seeds (e.g., 3) and modest scale
- Closed-source models (GPT-4) are used for training/evaluation in several settings, which may limit reproducibility and introduce model-specific biases.

---

> ### Author Rebuttal · Authors · 2026-03-31
>
> Thank you for the thoughtful feedback. We appreciate you advocating for a high degree of reproducibility and hope to address the mentioned concerns.
>
> ### Use of LLM judges
> We agree that reliance on LLM judges is worth mentioning and we now include it in the limitations. We ran a direct robustness check: we evaluated Exp. 4.1 with five judges (GPT-4o, GPT-4.1, GPT-4.1-mini, Claude Sonnet 4, Claude Haiku 4.5) over 50 generations for each of 108 samples. The hack metric is consistent across judges (mean pairwise Spearman 0.72; mean top-1 winner agreement 0.78). Beyond this, Secs. 4.2 and 4.3 use objective ground-truth evaluations entirely independent of any judge. Together, these increase our confidence that findings are robust to particular judge choice.
>
> ### Experiment scale
> We agree that some experiments use relatively few seeds. Due to compute constraints, we prioritized running all recontextualization variants, controls, and baselines in Sec 4.3 and 4.4, rather than additional seeds over a limited set of configurations. Our confidence relies on the consistency of the pattern across four distinct tasks, both proprietary and open-weight models, both judge-based and objective evaluations, and a wide range of prompt conditions. In addition, Sec 4.1 and 4.2 uses broader averaging, where recontextualization consistently improves correctness and reduces hacking. We therefore view the current results as meaningful evidence, while agreeing that larger-scale replications would strengthen the case further.
>
> ### Use of closed-source models
> We acknowledge that the GPT-4.1-mini experiments in Sec. 4.1 and 4.2 limits direct third-party reproducibility. For that reason, we also include open-weight experiments in Sec. 4.3 (Qwen3-4B) and Sec. 4.4 (Qwen3-8B-Base), which provide a reproducible foundation across two of the four settings; notably, Sec. 4.3 also uses objective evaluation rather than LLM judges. We included proprietary-model experiments to test whether the effect generalizes across model families and scales. The consistency of the results across open-weight and closed-source settings increases our confidence that the findings are not specific to one model or training setup. To further support reproducibility, we included our codebase as supplementary material, and plan to release it publicly so that our pipelines can be run on open-weight models across all settings.

---

> > ### Author Rebuttal · Reviewer_5Dov · 2026-04-02
> >
> > Thanks for the response and it resolves my concerns. I'll adjust my score accordingly.

---

### Official Review · Reviewer_qsat · 2026-03-14

**Soundness:** 3
**Presentation:** 2
**Significance:** 3
**Originality:** 3
**Overall Recommendation:** 4
**Confidence:** 2

**Summary:**

This paper proposes recontextualization, a training strategy aimed at mitigating specification gaming (e.g., reward hacking) in language models. The key idea is to generate model completions using prompts that discourage target misbehavior, and then recontextualize these same completions under prompts that permit or even encourage the misbehavior during training. The intuition is that if the model learns to behave correctly even when instructions allow misbehavior, it may become more robust to reward hacking arising from imperfect training signals. The paper evaluates recontextualization across several training environments prone to specification gaming and compares it with alternative interventions such as inoculation prompting, KL regularization, and reward hacking detection. Results suggest that recontextualization can significantly reduce specification gaming while maintaining competitive task performance.

**Compliance With Llm Reviewing Policy:**

Affirmed.

**Final Justification:**

My main concerns are addressed.

**Key Questions For Authors:**

- The paper provides an intuitive explanation for why recontextualization reduces specification gaming, but the underlying mechanism is not fully understood. For example, how sensitive is the method to the choice of generation prompts and training prompts? Are there theoretical or empirical insights explaining why certain prompt contrasts work better than others?
- The current experiments focus on training environments where the evaluation criteria can be gamed. However, in real-world scenarios, agents may actively search for exploits in the training signal. How robust is recontextualization under stronger adversarial optimization settings?

**Limitations:**

- The experiments focus on controlled environments where specification gaming is intentionally introduced. Further validation in more realistic or frontier training settings would be necessary to assess the practical impact of the method.
- Recontextualization is inherently off-policy, since completions are generated under one prompt context but used to train under another. This distribution shift may introduce instability or degrade performance in more complex training environments.

**Strengths And Weaknesses:**

Strengths
- The paper studies specification gaming and reward hacking, which is an important and timely topic in AI alignment and language model training.
- The proposed recontextualization strategy is conceptually novel and offers an interesting perspective on mitigating reward hacking without modifying the training specification itself.
- The empirical evaluation across multiple gaming-prone training environments demonstrates that recontextualization can reduce reward hacking while maintaining comparable model capability relative to baseline training methods.

Weaknesses
- Some sections require careful reading to understand the motivation and training pipeline, and the overall presentation could be streamlined to improve readability.
- The paper does not fully clarify several implementation choices, such as prompt construction strategies, sensitivity to prompt selection, and how robust the approach is to different training configurations.
- Limited discussion of adversarial settings. While the method aims to mitigate specification gaming, the paper does not thoroughly analyze potential adversarial behaviors that might emerge under this training paradigm.

---

> ### Author Rebuttal · Authors · 2026-03-31
>
> Thank you for the thoughtful review and for raising interesting questions and discussions. We also appreciate the presentation feedback and have revised the experimental sections to clarify the motivation, setup, and results.
>
> ### Underlying mechanism
> We agree that prompt sensitivity and the mechanism behind recontextualization are important open questions. Our current paper provides several empirical results on this issue: Section 4.1 compares generic and task-specific prompts, Section 4.3 studies five prompt variants, and Appendix G.3 examines how elicitation strength relates to training effectiveness. Taken together, these results suggest that recontextualization is not tied to a single hand-crafted prompt, but they also show that prompt choice matters and that we do not yet have a complete mechanistic account of when particular prompt contrasts work best. Our goal in this work is primarily to establish that the method is effective across several specification-gaming settings and to provide empirical observations that future work can build on mechanistically.
>
> This is also consistent with the current state of related work on inoculation prompting. Wichers et al. test multiple candidate inoculation prompts and propose an elicitation-based heuristic for selecting them, but offer only an intuitive explanation and note that the heuristic is not always reliable. Tan et al. ablate prompt paraphrases and learning dynamics, and suggest that inoculation works by making a trait less surprising thereby reducing pressure for broad model updates, while also noting that some results remain unexplained. Taken together, we think the field currently has useful behavioral evidence and candidate explanations, but a deeper mechanistic understanding remains ongoing work.
>
> ### Adversarial settings
> Our current environments are intentionally gameable to serve as controlled proxies for real training weaknesses: the core issue is an exploitable blind spot in the training signal. The key difference from real settings is that we know the exploit in advance, whereas in production a model may discover vulnerabilities before developers are aware of them. This is not unique to our setup: MacDiarmid et al. studies environments that were selected because they were known to be vulnerable, and gives the model information about those vulnerabilities to study the resulting dynamics and inoculation prompting. Our results suggest that recontextualization can still be useful in this broader regime because it can reduce hacking without prior knowledge of the exact exploit.
>
> We agree that at a larger scale or in more agentic settings, qualitatively new behaviors may emerge, including settings where a model actively searches for exploits in the training signal. In many realistic training failures, gaming emerges because the training signal gradually reinforces an exploitable blind spot. To our knowledge, models that terminally seek reward rather than task outcomes have not yet been widely observed and remain poorly understood. We see stronger adversarial settings as an important direction for future work.
>
> ### Off-policy concern
> We acknowledge that the off-policy nature of recontextualization is a potential drawback of the method, which we discuss in the limitations section and Appendix A. Empirically, we did not observe degraded task performance in the settings studied, but further validation in more complex environments remains important.

---

> > ### Author Rebuttal · Reviewer_qsat · 2026-04-02
> >
> > Thanks authors to the detailed rebuttal. My concerns are fully resolved and I will maintain my score.

---

### Official Review · Reviewer_aomf · 2026-03-16

**Soundness:** 3
**Presentation:** 4
**Significance:** 3
**Originality:** 3
**Overall Recommendation:** 5
**Confidence:** 3

**Summary:**

The paper considers the problem of discouraging models from misbehaving according to a specification by gaming it. They first collect data generation prompts that discourage a certain behavior. The model responses may be behaving or misbehaving. Regardless, at training, simply replace the data generation prompts with prompts that encourage the behavior. The idea is that the model will never be exposed to examples of query response pairs (discouraging misbehavior, model misbehaving) since every prompt will be encouraging misbehavior.

**Compliance With Llm Reviewing Policy:**

Affirmed.

**Key Questions For Authors:**

listed above

**Limitations:**

yes

**Strengths And Weaknesses:**

Strengths
1. The paper takes on the notoriously hard problem of reducing specification gaming.
2. The trick introduced in the paper is novel (to me) and is a clever way of ensuring there are no examples of misbehavior with a prompt requesting no misbehavior
3. There seem to be positive interactions with quality (in the cases where reward hacking and quality are orthogonal?)
4. The experimental evaluation is quite rigorous, covering a lot of baselines and problem settings where this would be of interest.

Weaknesses
1. Why does recontextualization improve performance in Figure 2b? This doesn't make sense to me since one is giving faulty instruction following data. It makes more sense in the MBPP example since correctness is defined differently for hack and correct.
2. It is very unclear to me what the training setup is (like what is the SFT/RL algorithm in all of the experiments). I see the first description of this in 4.3 after already seeing the experiments in 4.1-4.2.
3. The abstract/intro don't discuss the tradeoff between quality and hacking as much as it should. Please add more discussion there.
4. The above weakness especially applies to the negative IFEval result you show later, I find it bad its mentioned this late since a reader may leave with the impression there is no tradeoff until they get to the final table of the main body. I follow the stated reasoning for disagreement between AlpacaEval and IFEval, I do think this merits a lot more study.

---

> ### Author Rebuttal · Authors · 2026-03-31
>
> Thank you for the thoughtful feedback that allowed us to make several revisions to improve clarity and presentation.
>
> ### Performance in Figure 2b
> We do not interpret Fig. 2b as evidence of a meaningful quality gain from recontextualization. The increase is small and within noise relative to the baseline quality score of 9.24 (Fig. 2b; Table 4). However, responses that genuinely satisfy the task criteria without overfitting them score higher on both metrics than low-quality answers, so a flawed reward signal can still produce small improvements on quality as well. We now state this more clearly and clarify the relation of the quality and hack scores to the evaluation criteria in the main text and Appendix D.4.
>
> ### Training setups
> We agree that the per-experiment training setup was not sufficiently easy to track in the original draft. Concretely, Sec. 4.1 and Sec. 4.2 use single-step expert iteration, Sec. 4.3 uses 200 steps of GRPO, and Sec. 4.4 uses multi-step expert iteration. While these details were present, we have revised the experimental sections so that the training algorithm used in each setting is stated clearly from the outset.
>
> ### Quality/hacking tradeoff
> We agree that this tradeoff should be more salient. In line with our previous answer regarding Sec. 4.1, we revised the abstract to more accurately describe overfitting evaluation criteria at the expense of response quality. We also better highlight in the introduction that in settings such as sycophancy, undesirable behavior arises as a direct overfit to the very property we want models to optimize for, with no clear separability.
>
> ### IFEval results
> We agree these results should be mentioned earlier in the paper. We now highlight this point in the introduction and add clearer pointers to the relevant appendix discussion. Since the IFEval decline does not appear in all settings (such as Sec. 4.3), we are careful to present it as a conditional rather than an inevitable tradeoff. We agree that understanding when and how it occurs is an important direction for future work.

---

> > ### Author Rebuttal · Reviewer_aomf · 2026-04-06
> >
> > Thanks! I maintain my positive score.

---

### Decision · Program_Chairs · 2026-04-30

**Decision:**

Accept (regular)

**Comment:**

All reviewers are positive and converge after rebuttal. The paper presents a novel and practical idea for mitigating specification gaming with solid empirical support across settings, though some concerns remain about prompt sensitivity and realism. Overall, a worthwhile and timely contribution.